# Which Dark Personality Traits Could Predict Insomnia? The Mediated Effects of Perceived Stress and Ethical Judgments

**DOI:** 10.3390/bs13020122

**Published:** 2023-02-01

**Authors:** Seyed Hojjat Zamani Sani, Gianpiero Greco, Zahra Fathirezaie, Georgian Badicu, Mohammad Taghi Aghdasi, Kosar Abbaspour, Francesco Fischetti

**Affiliations:** 1Motor Behavior Faculty, Physical Education and Sport Sciences Faculty, University of Tabriz, Tabriz 51666, Iran; 2Department of Translational Biomedicine and Neuroscience (DiBraiN), University of Study of Bari, 70124 Bari, Italy; 3Department of Physical Education and Special Motricity, Faculty of Physical Education and Mountain Sports, Transilvania University of Braşov, 500068 Braşov, Romania

**Keywords:** dark personality traits, insomnia, perceived stress, ethical judgments

## Abstract

This study aimed to investigate the relationship between dark personality traits and insomnia by considering the mediated effects of perceived stress and ethical judgments. This descriptive and correlational study was conducted with 464 university athlete students from individual and team sports. Dark Triad Scale (DTS), Insomnia Severity Index (ISI), Perceived Stress Scale (PSS), and Moral Content Judgment in Sport Questionnaire (MCJSQ) were used. Significant correlation coefficients were observed between the dark personality traits and other variables. Regression analysis showed that psychopathy (about 19%) and ethical judgments (about 16%) could predict insomnia. It was shown that among dark personality traits, psychopathy along with ethical judgments could predict insomnia.

## 1. Introduction

Sleep is known to be an essential element for recovery and its quality is recognized by many athletes as the most important way to improve their lifestyle [1]. Research has shown that better sleep quality affects homeostatic, neurological [2], endocrine [3], and immune [4] regulations. It is also an important element that helps the athlete to improve his physical and mental well-being [5]. Plenty of past research has detailed that exercise execution is negatively influenced following sleep loss [6]. By contrast, the quality of sleep has been highlighted as decisive for an athlete’s ability to train, improve performance, and prevent injury and recovery [7,8]. Research has also shown that sleep has a direct relationship with mood [9] and sports performance [10]. Recently, studies have strengthened the relationship between sleep quality, mood, and sports performance [11,12,13,14]. From a psychological perspective, scientists have placed great emphasis on examining relationships between sleep quality and personality factors [15]. However, there is a dearth of research on the role of psychological variables and how these relate to sleep quality. Recently, the study of dark aspects of personality has become popular among researchers. Sport psychology recognizes the value of dark traits in both research and practice [16]. Paulhus and Williams [17] defined the dark triad (DT) as a set of three personality traits, such that “individuals with these traits share a tendency to be callous, selfish, and malevolent in their interpersonal dealings”. Machiavellianism is a manipulative personality. Manipulative personalities are characterized by a lack of empathy, low affect, possessing an unconventional view of morality—a willingness to manipulate, lie to, and exploit others—and focusing exclusively on their own goals/agenda, not those of others [18]. Narcissism has facets from its clinical variant: grandiosity, entitlement, dominance, and superiority. Moreover, psychopathy has been described as impulsivity and thrill-seeking combined with low empathy and anxiety [19]. Psychopaths are antagonistic and believe in their own superiority, and have a tendency towards self-promotion [20,21]. Although the dark dimensions of personality include negative aspects, they can also have a positive effect on sport [22,23] and social success, for example in specific occupations [24,25]. Past research [26,27] has shown relationships between some dark personality traits and sleep disturbances in different groups. Therefore, in the first hypothesis of the present research, we expected to observe a significant relationship between dark personality traits and insomnia in athletes (HYP1).

It seems that multiple factors, including personality and environmental characteristics, could affect this relationship. Therefore, other dimensions of the research literature on sleep quality are considered. Perceived stress is another important issue in sports that has received much attention and has been evaluated with regards sleep quality and personality. Andersen and Williams [28], with the same line of conceptualization, proposed a ‘stress-athletic model’, which contends that in the stress appraisal process, athletes’ perceived stress is influenced by certain factors, such as personality. The results of past research have pointed out the relationship between narcissism and stress [29,30]. Therefore, in the second hypothesis of the current research, we assumed that there would probably be a significant relationship between the dark characteristics of athletes’ personalities and perceived stress (HYP2).

In addition, previous studies have shown that many athletes who suffer from poor sleep may have negative stress [31]. In one longitudinal study, day-level stress was positively associated with perseverative cognitions (worry), which were significantly related to impaired sleep [32]. Although it can be said that stress predicts poor sleep [33], taken together, these studies attest to a bi-directionality between stress and sleep that may proceed in a vicious cycle. Importantly, not all people exposed to stressors experience disturbed sleep. Some of them have the ability to overcome adversity and adapt to new circumstances positively [34]. This adaptive ability in some people seems to be related to their psychological and personality traits; more research is needed in this regard. Therefore, based on the research background, the third hypothesis of this research stated that there is probably a significant relationship between athletes’ perceived stress and their insomnia, and maybe the perceived stress along with other variables can predict athletes’ insomnia (HYP3).

Research in sporting [35] and academic [36] situations has shown that stress also affects ethical judgments. According to research, personality characteristics affect various aspects of behavior; for example, there is evidence that narcissism is related to ethical judgment [37,38]. However, Schepers [39] showed that there is no significant relationship between ethical judgment factors and Machiavellianism among MBA students. In addition, Doty [40] reported contradictory results and cited that moral development appears to be a lifelong, holistic process, which is primarily influenced by contextual variables throughout a person’s life. However, if sport is part of a person’s life, then the sports experience should have the power to influence his or her character development (hopefully in a positive way). Therefore, hypotheses number four to six were formulated like this:

We expected that perceived stress and ethical judgments were associated (HYP4).

We also predicted that the dark personality traits of the athletes are probably related to their moral judgments (HYP5).

Finally, the sixth hypothesis stated that moral judgments probably have a relationship with insomnia and can predict it along with other variables (HYP6).

## 2. Materials and Methods

### 2.1. Subjects and Design

The type of the present study was cross-sectional and retrospective. The study population was Iranian university athlete students. There were 464 people (57% female and 43% male) from various Iranian universities who were involved in individual and team sports. They were selected by cluster sampling and voluntarily participated in this study. Of the 464 participants (mean age = 23.15 years; SD = 1.63; 47.9% males) who took part in the study, 70.7% were team sport athletes and 29.3% were individual sport athletes. According to their self-reports, all the athletes undertook specialized training at least three days a week in the three months before the start of the competition.

### 2.2. Procedure

Students participating in university sports competitions from Tehran, Shiraz, and Tabriz University (Tehran, Shiraz, and Tabriz, Iran) were approached to participate in the present study. They were fully informed about the aims of the study and the anonymous data gathering and data elaboration. Thereafter, participants signed their written informed consent. Next, participants completed a booklet of self-rating covering sociodemographic information, DTS, ISI, PSS, MCJSQ, and RCI-10 questionnaires.

### 2.3. Instruments

Demographic characteristics (age, sex, the field of study, and sport) were collected from all subjects.

Dark triad scale (DTS): The 12-item Dark Triad Dirty Dozen Scale [41] was used to assess Machiavellianism (e.g., “I tend to exploit others towards my own end”), psychopathy (e.g., “I tend to be callous or insensitive”), and narcissism (e.g., “I tend to seek prestige or status”). In order to create scores for Machiavellianism (CRS = 0.90), psychopathy (CRS = 0.85), and narcissism (CRS = 0.92), items (1 = strongly disagree, 9 = strongly agree) for each scale were averaged together [42].

Insomnia Severity Index (ISI): Insomnia Severity Index was used to assess insomnia. ISI, which has seven items, is a brief self-reported questionnaire. This questionnaire examines the sleep patterns in the two weeks immediately preceding the administration of the test. The participant’s perception of both nocturnal and diurnal symptoms is examined by the items, rated on a scale from 0 (less severe) to 4 (more severe). The scores of the seven items, which range from 0 to 28, are added to generate the total score [43]. A cut-off score of 10 has 86.1% sensitivity and 87.7% specificity for detecting insomnia cases in community samples [43].

Perceived Stress Scale (PSS): This questionnaire was developed by Sheldon Cohen in the year 1994. The Perceived Stress Scale (PSS) is the most commonly utilized psychological tool for assessing stress perception. The scale also includes a variety of direct questions about current levels of experienced stress. The PSS is available in three versions: the original 14-item scale (PSS-14), a 10-item scale (PSS 10), and a 4-item scale (PSS-4). In this study, the PSS-10 version was used, which has six items that are negatively stated and four items that are positively stated. Through this, it measures how stressful a person’s life is. The reliability of PSS is 0.85 and its validity ranges from 0.52 to 0.76. The PSS has a range of scores between 0 and 40 and a lower score indicates less stress [44].

Moral Content Judgment in Sport Questionnaire (MCJSQ): In the present study, the last 24-item structure was utilized. The instrument began with the statement: “Do I believe that my actions in sport are characterized by…?” This statement was followed by 25 items related to the five constructs of the moral content elements normative order (e.g., “…interest in the opponents when the latter are in danger.”), egoistic utilitarianism/consequences (e.g., “…a wish for reward.”), social utilitarianism/consequences (e.g., “…an interest in the positive consequences for my team.”), harmony-serving consequences (e.g., “…courage and nerve.”), and fairness (e.g., “respect to the opponent.”). The participants were asked to reply on a 9-point Likert-type scale ranging from 1 (strongly disagree) to 9 (strongly agree). This scale was chosen because, in the previous studies, which utilized a 5-point Likert-type scale, this method was used to avoid the accumulation of values close to scales 4 and 5 [45].

### 2.4. Data Analysis

A series of Pearson’s correlations were performed to calculate the associations between dark personality traits, insomnia, perceived stress, and moral judgment.

A multiple linear regression analysis was performed to predict insomnia; predictors were dark triad personality traits, perceived stress, and moral judgment. All statistical analyses were computed utilizing SPSS^®^ 25.0 (IBM Corporation, Armonk, NY, USA).

## 3. Results

As shown below, among the features of dark triad personality characteristics, narcissism had a higher score (Table 1).

Only psychopathy was positively related to insomnia. Other personality features did not relate to insomnia. In addition, perceived stress and ethical judgment had a positive significant relationship with insomnia (Table 1).

All relationships between dark personality traits were positive and significant. In addition, among the three personality traits, only narcissism positively related to ethical judgment and perceived stress (Table 1).

In addition to the above, perceived stress was positively related to ethical judgment (Table 1).

Descriptive and correlational statistics of study variables are shown in Table 1.

Multiple linear regression with insomnia as the dependent variable, and personality traits, ethical judgment, and perceived stress as predictors, showed that the model related to insomnia prediction is significant with the study variables (Table 2, Table 3 and Table 4).

It was shown that psychopathy and ethical judgments could predict insomnia.

## 4. Discussion

This study aimed to investigate the relationship between dark personality traits (Machiavellianism, psychopathy, and narcissism) and insomnia, perceived stress, and ethical judgments. A total of 464 athlete students participated, and DTS, ISI, PSS, and MCJSQ were completed. Results showed that there is a significant relationship between insomnia and psychopathy, ethical judgment, and perceived stress. In addition, perceived stress had a relation to ethical judgment. The regression analysis model showed that insomnia could be predicted by psychopathy and ethical judgment, but only in small quantities (6%).

Our results provide evidence that specific aspects of the dark triad personality are associated with symptoms of insomnia. Therefore, regarding the first hypothesis, only the psychopathy trait was confirmed, and no relationship was observed between the other two personality factors (narcissism and Machiavellianism) and insomnia. Given that individuals high in psychopathy indicate a lack of both emotional control and adaptive coping strategies [46], one suggested pathway may be that insomnia symptoms present amongst those high in psychopathy may be due to deficits in emotion regulation, which then serves to accentuate negatively toned cognitive activity resulting in poor sleep. These outcomes are partly in line with recent research conducted by Akram et al. [26]. They demonstrated that Machiavellianism and psychopathy, but not narcissism, were related to increased reports of sleep disturbance [26]. The relationship between insomnia and dark personality traits seems to be explained by adverse cognitive–emotional processes (i.e., worry, rumination, poor coping, and diminished emotion regulation). To describe precisely, a key characteristic of insomnia is the presence of negatively toned cognitive activity, mostly in terms of worry and rumination [47]. Based on the correlation results obtained in the athletic community of the present study, as well as the study of Sabouri et al. [27], it can be concluded that among personality traits, psychopathic features should be considered concerning insomnia symptoms.

The results also showed that insomnia and stress as well as insomnia and ethical judgments are positively and significantly related. These stress-related observations were in line with the results of Palagini et al. [46], Pillai et al. [48], Drake et al. [49], Morin et al. [50], and Cuadros et al. [51]. Pillai et al. [48] cited that stress exposure is a critical predictor of insomnia onset. This means that people who have more stress in life or perceive events with more stress have a lower quality of sleep. According to Morin et al. [50], insomniacs rated both the impact of daily minor stressors and the intensity of major negative life events higher than good sleepers. In addition, Palagini et al. [46] suggested that some people who have both immune system problems and symptoms of insomnia are more likely to perceive challenging events as stressful. In general, we can say that stressful events and daily stress create psychological and organic symptoms that impair quality of life and sleep [51]. In addition, research has shown that the quality of ethical judgment dilemmas is influenced by a long lack of sleep [52]. In terms of the structural function of the brain, the prefrontal cortex, which plays a prominent role in the formation of ethical decisions, is particularly sensitive to sleep loss. Therefore, the relation between ethical judgments and insomnia can be explained biologically. Therefore, the third hypothesis was partially confirmed and the sixth hypothesis was fully confirmed. Although perceived stress and moral judgment were related to insomnia, moral judgment was able to predict insomnia along with other variables.

Our results related to dark personality traits provide evidence that among its subscales, there was a significant positive relationship between narcissism and moral judgment as well as perceived stress. Results concerning moral judgment were in line with the results of Saculla et al. [53], who researched the relationship between narcissism and moral judgment development. In this way, neuroimaging studies on narcissism have highlighted that the presence of harm, pain, suffering, victimhood, and dominance might evoke a vicarious source of pleasurable and rewarding experiences [54]. By contrast, results about perceived stress were in line with the results of Kajonius et al. [29] and Gore et al. [30]. The results imply that a vulnerable narcissistic personality disposition relates to increased affects from stressful circumstances, while a grandiose personality may thrive even in stressful environments [30]. It can also be stated that persons with vulnerable narcissistic traits more readily perceive situations as threatening to the self and thus as stressful, rather than seeking out a lifestyle that is more stressful [29]. Therefore, the second and fifth hypotheses were partially confirmed.

This research pointed out that there was a positive and significant relationship between perceived stress and moral judgment. The results of the present study were consistent with previous studies [35,36]. In addition, Youssef et al. [55] found that activation of the stress response, which was specific to personal moral dilemmas, happens because of a reduction in utilitarian responses. Therefore, the fourth hypothesis of the current research was confirmed.

In the study, we performed linear regression analysis to predict insomnia by other variables. This analysis pointed out that among the variables, psychopathy and moral judgment were able to predict insomnia. By combining the variables, it was shown that some variables, though related to insomnia alone, were not able to predict insomnia along with other variables. Therefore, psychopathy and moral judgment positively could predict insomnia. Therefore, the evidence again confirmed the first and third hypotheses of the research. Nonetheless, its numerical value was very small.

## 5. Conclusions

In this study, psychopathy showed a significant relationship with insomnia, confirming previous studies. However, it must be acknowledged that the extent to which personality factors mediate and encourage the onset of insomnia is poorly understood [56] and there is conflicting information regarding which personality traits are associated with sleep disturbance/insomnia [57]. In particular, this issue can be affected by individual characteristics such as age, gender, certain subcultural factors, etc. Therefore, the generalizability of findings should be considered in light of the effect of the communities’ cultural features as well as the characteristics of the individuals. These relations were confirmed among a group of athletes; however, this research should also be conducted for other specific groups, taking into account individual characteristics. Regarding the relationship between different personality traits and insomnia, psychopathy directly and principally affected insomnia. Narcissism and Machiavellianism both indirectly affected insomnia, with narcissism being the second most important factor and Machiavellianism the third most important factor.

## Figures and Tables

**Table 1 behavsci-13-00122-t001:** Correlation coefficients of study variables.

	Mean (SD)	1	2	3	4	5
Dark Triad (overall score)	29.38 (5.88)					
1. Machiavellianism	7.20 (2.98)					
2. Psychopathy	7.96 (2.38)	0.348 **				
3. Narcissism	14.27 (2.86)	0.250 **	0.172 **			
4. Insomnia	14.75 (4.26)	0.043	0.180 **	0.069		
5. Ethical Judgment	90.11 (11.99)	−0.082	−0.057	0.253 **	0.168 **	
6. Perceived Stress	30.37 (3.67)	0.064	0.029	0.243 **	0.135 **	0.317 **

** *p* ≤ 0.01.

**Table 2 behavsci-13-00122-t002:** Model parameters for the prediction of insomnia.

Model	R	R Square	Adjusted R Square	Std. Error of the Estimate	Durbin-Watson
	0.266	0.071	0.061	4.131	1.989

**Table 3 behavsci-13-00122-t003:** The third fit model parameters for the prediction of insomnia.

	Sum of Squares	df	Mean Square	F	p
Regression	596.64	5	119.328	6.9989	0.000
Residual	7819.54	458	17.07		
Total	8416.18	463			

**Table 4 behavsci-13-00122-t004:** Coefficients of the third model for the prediction of insomnia.

	**Unstandardized** **Coefficients**		**Standardized** **Coefficients**	**t**	**p**
	**B**	**Std. Error**	**Beta**		
(Constant)	4.539	2.025		2.242	0.025
Machiavellianism	−0.017	0.071	−0.012	−0.239	0.812
Psychopathy	0.347	0.086	0.194	4.019	0.000
Narcissism	−0.033	0.073	−0.22	−0.0453	0.651
Perceived Stress	0.100	0.056	0.086	1.788	0.075
Ethical Content Judgment	0.056	0.017	0.156	3.186	0.002

## Data Availability

Not applicable.

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
