# Peer review of "Which Dark Personality Traits Could Predict Insomnia? The Mediated Effects of Perceived Stress and Ethical Judgments"

_behavsci, 2023, doi:10.3390/bs13020122_

Round 1

Reviewer 1 Report

The study is relevant to the field and its results contribute to addressing a gap in the study of psychological traits and insomnia in athletes.

The introduction gives an overview of the research findings in the area of study, focusing on the variables the authors are considering, i.e. the influence of dark personality traits, perceived stress and moral judgments on insomnia. 

Objectives: Although the six hypotheses were listed, only the second objective was mentioned, but not the first.  Page 2, lines 75-76 is described the second objective "So, the second aim of this study was to investigate the relationship between personality characteristics, perceived stress, and insomnia." The first aim was described in the abstract. Usually the objectives are mentioned and then the hypotheses.

Results: From Table 04 which shows the results of the Multiple linear regression it does not appear that the interaction between the variables was tested. The results of this table indicate that both psychopathy and ethical judgments predict insomnia, but not the interaction of these variables as described in rows 199 and 200, as well as in the abstract rows 20 and 21. 

The Discussion is well described and supported.

Reviewer 2 Report

I had the opportunity to read & review a very interesting study about dark personality traits and their prediction of insomnia. I have just a few comments for the authors:

Introduction:

- Lines 53-55, 63-66, 75-80, 90-96: maybe it would be better to move the aims & hypotheses to the end of the chapter and prepare a subsection with all the aims and goals.

Materials & Methods:

- Lines 99-100: I am sorry, but I don’t think I understand this sentence.

- Lines 111-114: I don’t think this is necessary as you already have this information in the Institutional Review Board Statement.

Results:

- Lines 160-163: I believe it would be better to move this text to the Materials & Methods (Subjects & Design).

- Lines 163-177: I don’t think it is necessary to include so many subtitles – the text would be more cohesive without them.

- Table 1: I believe there is a mistake in the coefficient between variables 3 and 4.

- Lines 199-200: I am sorry, but I don’t find the statistical proof for this statement, could you please add it?

Discussion:

- Line 211: are you sure you can confirm this hypothesis? You only confirmed the correlation between psychopathy and insomnia, and not with other two dark traits!

Round 2

Reviewer 2 Report

Thank you for the review, it has been done in a very good manner. I just have, again, a comment regarding the already mentioned coefficient in the Table 1: I am pretty sure that the coefficient "0.69" would be significant as it is quite high (for example: coeff. between variables 1 and 2 is 0.348 and it is strongly significant) - so I believe there is a typo and the right coefficient is 0.069 (or, abbreviated, .069).
